# Quality Assessment of Internet Information Regarding Periodontitis in Persons Living with HIV

**DOI:** 10.3390/ijerph21070857

**Published:** 2024-06-29

**Authors:** Hester Groenewegen, Arjan Vissink, Fred K. L. Spijkervet, Wouter F. W. Bierman, Konstantina Delli

**Affiliations:** 1Department of Oral and Maxillofacial Surgery, University Medical Center Groningen, University of Groningen, P.O. Box 30.001, 9700 RB Groningen, The Netherlands; f.k.l.spijkervet@umcg.nl (F.K.L.S.); k.delli@umcg.nl (K.D.); 2Department of Internal Medicine, Division of Infectious Diseases, University Medical Center Groningen, University of Groningen, P.O. Box 30.001, 9700 RB Groningen, The Netherlands; w.f.w.bierman@umcg.nl

**Keywords:** HIV, periodontitis, readability, quality, websites, internet information

## Abstract

The Internet is the most used source of HIV information second to information received from healthcare professionals. The aim of this study was to assess the quality of Internet information about periodontitis in people living with HIV (PLWH). An Internet search was performed on 18 April 2024 using the search terms “Periodontitis”, “Periodontal disease”, and “Gum disease” in combination with “HIV” in the most popular search engines (Google™, Bing™, and YAHOO!^®^). The first 20 results from each search term engine were pooled for analysis. Quality was assessed by JAMA benchmarks. Readability was assessed using the Flesch reading ease score (FRES). Origin of the site, type of author, and information details were also recorded. The quality of Internet information about periodontitis in PLWH varied. The mean JAMA score was 2.81 (*SD* = 1.0). The websites were generally fairly difficult to read (mean FRES = 57.1, *SD* = 15.0). Most websites provided some advice about self-treatment of oral problems, accompanied by a strong recommendation to seek professional dental care. In conclusion, advanced reading skills on periodontitis in PLWH were required and quality features were mostly not provided. Therefore, healthcare professionals should be actively involved in developing high-quality information resources and direct patients to evidence-based materials on the Internet.

## 1. Introduction

Human immunodeficiency virus (HIV) infection has evolved to become a chronic disease [1,2]. In 2022, approximately 39.0 million worldwide (33.1 million–45.7 million) people were living with HIV [3], including early stages to more advanced stages of HIV, i.e., acquired immunodeficiency syndrome (AIDS). The life expectancy and quality of life in people living with HIV (PLWH) is less determined by HIV but rather by age-related diseases and comorbidities [4]. As a result, the comorbidity-free years of life expectancy in PLWH have been shown to be lower than in people without HIV [5]. A frequently diagnosed comorbidity in PLWH is periodontitis. Periodontitis is a microbe-associated inflammation that can result in tooth loss [6] and is associated with several systemic diseases, such as cardiovascular diseases, auto-immune diseases, and diabetes mellitus [7,8,9] The prevalence of chronic periodontitis in the general population is approximately 30%. Severe generalized periodontitis is present in 5–15% of the population worldwide [10,11]. In PLWH, the prevalence of periodontitis ranges from 30% up to 100%, depending on the type of HIV treatment used and the classification applied to diagnose periodontitis [12,13,14].

Recently, we reported that 46% of PLWH did not inform their dentist about their HIV status [14]. We assumed that HIV-related stigma, defined as a set of negative and often unfair beliefs that a society or group of people have about something, might play a factor in not informing dentists. HIV-related stigma is still a present problem and could lead to discrimination [15]. This discrimination can have a negative impact on the quality of their oral health care and prevent them from accessing dental services [16]. This can lead to a significant deterioration of oral health-related quality of life, in particular periodontitis, as studies have shown [17,18].

For PLWH, the Internet is the most used source of HIV information, second to information obtained from their healthcare professional [19,20]. Particularly in settings lacking robust data on people’s health information needs, analyzing the population’s reliance on search engines to fulfill those needs can yield valuable insights [21]. The Internet is already available for more than half of all people worldwide (available online: https://www.statista.com, accessed on 18 April 2024), making information, including medical information, more obtainable than ever before [22,23]. The widespread availability of Internet health information may have an impact on the patient–physician relationship, since trustworthy online medical information can encourage patients in the management of their health and be of aid in seeking treatment-related health information. Internet health information can also improve patient compliance [24,25,26].

To date, the quantitative and qualitative characteristics of the online information for PLWH about periodontitis are unknown. Therefore, the aim of this study was to assess the quality of available Internet information about periodontitis in PLWH.

## 2. Methods

### 2.1. Study Design

This was a cross-sectional study of Internet sites relevant to periodontal disease and HIV. An Internet search was performed to identify available websites dealing with information about HIV and periodontitis. Ethical approval was not required due to the nature of this study.

### 2.2. Search Method

An Internet search was performed on 18 April 2024, using the three most popular search engines according to Stat counter (www.statcounter.com, accessed on 18 April 2024), a web traffic analysis website, i.e., Google™ (www.google.com, accessed on 18 April 2024), Bing™ (www.Bing.com, accessed on 18 April 2024), and YAHOO!^®^ (www.yahoo.com, accessed on 18 April 2024). We selected the keywords according to Google Trends (https://trends.google.com/trends/?geo=EN, accessed on 18 April 2024) [27]. The search terms “Periodontitis”, “Periodontal disease”, and “Gum disease” were found as the most frequently used search terms when seeking information about periodontitis. Regarding the search term for HIV, “HIV” was the only frequently used search term found. The search terms “Periodontitis”, “Periodontal disease”, and “Gum disease” were entered separately, each combined with “HIV”. This was performed separately in all 3 search engines, resulting in a total of 9 searches. Search histories and Internet caches (including cookies) were cleared before every search. To prevent any potential biases that could arise from previous searches, we browsed in private mode. We compiled the first 20 results from all 9 searches. Exclusion criteria for websites were promotional product sites, discussion groups, scientific articles, non-operative sources, video feeds, sites with denied direct access through password requirement, non-English language domains, online medical dictionaries, social forums and social media websites, and irrelevant websites. Duplicate websites were removed. Websites were identified and examined by one reviewer (H.G.). In case of doubt during the assessment, an independent reviewer was consulted (K.D.).

### 2.3. Quality Assessment

To evaluate the quality of the websites, the following tools were used: JAMA benchmarks and the Flesch reading ease score (FRES) [28,29].

#### 2.3.1. JAMA Benchmarks

The JAMA criteria are designed to assess the quality of medical information on the website. The four website parameters assessed are as follows: (i) authorship, including authors, contributors, their affiliations, and credentials; (ii) attribution, including references, sources for all content, and all relevant copyright information; (iii) disclosure, including ownership of the website; and (iv) currency, i.e., date of published data. In the case of a positive answer for each parameter, a score of one point is given, while, in the case the criterion is not fulfilled, zero points are given. The JAMA score ranges from 0 (no criteria fulfilled) to 4 points (all 4 criteria fulfilled) [28].

#### 2.3.2. Flesch Reading Ease Score (FRES)

FRES is one of the oldest textual difficulty measures [29]. FRES is based on a formula that intends to measure the grade reading level based on the average sentence length and average number of syllables per word. Specifically, a part of text from the website of 200–500 words must be copy-pasted into an online FRES calculator program (http://www.readabilityformulas.com/free-readability-formula-tests.php, accessed on 24 April 2024). The outcome is a number ranging from 0 to 100, where a higher score indicates a text that is easy to read. The result obtained from this test corresponds to a respective US educational level. For example, scores between 90 and 100 are considered easily understandable by an average fifth grade student (classified as very easy). The score between 60 and 70 is largely considered acceptable and considered easily understood by 8th and 9th graders (classified as standard). Understanding texts with a score up to 30 is considered representative of a higher education level such as college or university graduate level (classified as very difficult).

#### 2.3.3. Other Criteria

In addition to the abovementioned quality criteria, we also assessed:Type of website, i.e., website preliminary dedicated to provide information about HIV that also presented some information about periodontitis or website on oral health (e.g., from a dental practice) that also included some information about HIV;Audience target, i.e., PLWH, healthcare workers, general population;Origin of the websites, i.e., government, site dental practice/care, patient information groups, HIV organization, news agencies;Type of clinical information presented on the website about periodontitis related to HIV, i.e., information about etiology of periodontal disease, symptoms of periodontal disease, or treatment options;Type of oral or periodontal problems discussed related to HIV, e.g., candida infection, Kaposi’s sarcoma, or periodontitis;Correctness of information.

### 2.4. Statistical Analysis

Statistical data processing was performed using SPSS software, version 28 (SPSS, Chicago, IL, USA). For determining the distribution of the data, the Shapiro–Wilk test for small sample sizes (<50 samples) and graphical interpretation of Q-Q plots were used. Regarding FRES, the Shapiro–Wilk test provided a *p* = 0.666 and, in combination with the Q-Q plots, it was concluded that a normal distribution was followed. For the JAMA score, the Shapiro–Wilk test yielded a *p* = 0.05 and, in combination with the Q-Q plots, it was concluded that a non-normal distribution was followed. Independent sample t-test, one way analysis of variance (ANOVA), and Chi-squared test were used where appropriate to test differences between the groups, i.e., FRES in the different JAMA score groups, and JAMA score in websites with different origins. For all tests, a *p* value < 0.05 was considered significant.

## 3. Results

### 3.1. Search Results

The initial searches using the three search engines yielded a total of 20,208,000 hits, with the most hits on search engines Google™ for “Gum diseases” and “HIV” (9,610,000) and the smallest number of hits on BING™ for “Periodontitis” and “HIV” (398,000) (Appendix A). By selecting the first 20 web sites found by each search, we originally yielded 180 search results for relevance analysis. Based on the eligibility criteria, we afterwards excluded 123 scientific articles, 7 medical guidelines, 2 not relevant sites, and 1 not operative site. After removing duplicates, 21 sites were finally included for qualitative and quantitative analysis (Figure 1).

### 3.2. Quality of the Included Websites

#### 3.2.1. JAMA Criteria

The overall mean score of the JAMA criteria was 2.81 (*SD* = 1.0). Only seven sites (33.3%) scored the maximum score of 4 and the origin of these websites were news agencies. In 61.9% of the sites (*N* = 13), the author was anonymous (parameter: authorship, Table 1). Of the eight websites with a known author, three were written by dentists and five by medical doctors. Of these eight websites, six were managed by news agencies, two by HIV organizations, and one by a non-profit organization. Eight sites of the total twenty-one included sites were written for the general population. Twelve of these twenty-one sites were meant specifically for PLWH and one for health workers (Table 1). The JAMA score was not significantly different between the websites meant for various target groups (*p* = 0.231).

Eighteen (85.7%) of the websites mentioned a date of online publication (parameter: currency). In five of these sites (27.8 %), the information was not older than 1 year. Interestingly, 11 websites mentioned references, sources for the content, and relevant copyright information (parameter: attribution; Table 1 and Table 2). All of these 11 sites mentioned references, and published a date on their site to indicate when the last update took place (parameter: currency). Of those 11 sites mentioned, references with parameter currency were in four of these sites and the information was updated during the last 12 months. Disclosures of the website, e.g., the potential conflict of interest arising from ownership, sponsorship, or advertising, were addressed in all sites (parameter: disclosure; Table 1 and Table 2).

#### 3.2.2. Flesch Reading Ease Score

FRES had a mean score of 57.1 (*SD* = 15.0) and ranged between 23 and 89, thus websites were deemed appropriate for grade level 10–12 (i.e., students in this grade are usually 15–18 years old). Six sites were classified as a standard reading level, four sites were classified as difficult, and one site as very difficult to read (Table 2). Only one website was classified as easy to read (FRES = 89), which also provided additional pictures. FRES was not significantly different between the various JAMA score categories (*p* = 0.472; Figure 2). The seven sites with JAMA score 4 were not easy to read according to FRES (mean FRES = 53, *SD* = 12.0). Interestingly, the mean FRES of the two sites with JAMA score 1, and thus of lower quality, were the easiest to read (mean FRES = 72.5, *SD* = 23.3; Figure 2). Eight of the twelve sites that were specifically written for PLWH were easy to read or standard to read according to FRES. Out of the other three sites specifically written for PLWH, three were fairly difficult to read and one was difficult to read according to FRES. At the same time, the website for health workers was fairly difficult to read according to FRES.

#### 3.2.3. Additional Information about Websites

As far as the origin of the websites was concerned, five websites were attributed to government, five were from dental organizations or companies specializing in dental care products, and five were from HIV organizations (Table 2). The JAMA score was associated with the origin of the websites (*p* = 0.21), with sites supported by new agencies having the highest JAMA score (median = 4, IQR = 3–4). Specifically, of the four sites from news agencies, two sites had a JAMA score of 4 and two had a JAMA score of 3.

Regarding the information presented, the advice given on the websites about treatment of oral diseases or periodontitis and recommendations when to visit an oral healthcare professional was correct. Eight of the twelve sites that were written specifically for PLWH provided advice about oral care or periodontal treatment and recommendations to visit an oral healthcare professional at first signs of inflamed, red, or bleeding gums or mouth lesions and offered tips on how to prevent them. However, most websites did not go into details about the treatment and did not clearly explain the possible association of periodontitis and HIV.

Additionally, three sites presented general information about HIV, with some specific information about periodontal diseases, and five sites provided general information about periodontal diseases, with some information about the effect of HIV on periodontal diseases. Thirteen presented general information about HIV and periodontal diseases. Fourteen sites described information about all kind of periodontal and oral diseases related to HIV infection, like candida infection or oral warts, and only five sites presented information only about periodontal diseases, like gingivitis or periodontitis (Table 2). Furthermore, it appeared that the two websites with the most recent update, <4 months, provided specific information about periodontal or oral diseases. The sites where the date of creation or update was missing provided information about periodontal diseases, but it was not possible for PLWH to identify whether it was recent or outdated and thus how reliable this information was.

## 4. Discussion

The Internet information for PLWH on periodontitis varied greatly in quality. To the best of our knowledge, no data have been published yet about the quality of the Internet information regarding periodontitis in PLWH. Although several studies have been published about Internet information or periodontitis in general [30,31,32] and about the quality in general of Internet information specifically meant for PLWH [33,34], no quality assessment has been conducted regarding the link between periodontal disease and PLWH. In this study, the quality of websites providing information about HIV and periodontitis was assessed by using the JAMA benchmark and FRES.

Regarding the JAMA benchmark, in our study, we found only seven sites with the maximum score of 4. These results differ from the studies on the quality of Internet information about periodontitis and oral cancer, which have found even lower JAMA scores. Specifically, the study of Kanmaz et al. (2021), which assessed the quality of online information on periodontitis, found no webpages fulfilling all four of the JAMA criteria [32]. Similarly, the study of Patton et al. (2014), which assessed the quality of online information on dental care for patients with cancer, found only 1 website out of the 32 included ones fulfilling all four JAMA criteria [35]. The findings from the studies that only assessed the quality of Internet information about periodontitis and adult orthodontics are consistent with our results, also showing low JAMA scores, as in our study [31,36]. Regarding the disclosure and attribution parameters of JAMA, our results are in agreement with the studies that investigated the quality of Internet information for oral cancer and lingual orthodontics, respectively, and that also found that the principle of disclosure was frequently adhered to while the principle of attribution was poorly adhered to [35,37].

Because a JAMA score of 3 or above is considered of high quality, a lower JAMA score reflects a less transparent and possibly unreliable website [28]. Almost half of the websites eligible in this study had a JAMA score below the expected standard required for patient information sources, qualifying these websites as less transparent and possibly unreliable. In addition, the JAMA score was found not to be associated with the origin of the website. However, websites managed by news agencies had higher JAMA scores, which may be explained by the fact these journalistic websites routinely present the name of the author and date of publication or update and thus adhere to the JAMA requirements.

The FRES reported in our study showed that the online information concerning HIV and periodontitis is generally fairly difficult to very difficult to read. This is a common observation, as studies on the quality of Internet information about xerostomia and the standards of online oral hygiene instructions for patients with fixed orthodontic appliances also showed that web information was generally written in a fairly difficult to very difficult understandable form [38,39]. Organizations such as the American Medical Association and the National Institutes of Health recommend that the readability of patient education materials should be no higher than a sixth grade reading level (approximately 11 years old) [40]. This recommendation has been drawn, because information that is very difficult to read potentially limits its accessibility and thereby hampers health literacy (which is associated with overall health) [41]. Thus, there is a general need to make the information on websites easier to understand. A reason for a general high FRES might be that online websites are often written by physicians or scientists who use complex medical terminology that is not well known or easy to understand by the general public [42]. Involving patients in the development process of the website through patient groups could help narrow the communication gap between doctors and patients [42].

As mentioned, the Internet is the most preferred source of HIV information, after information received from a healthcare professional [19,20]. In the study of Threats et al., the main sources of HIV information on the web were consumer health information websites, government health information websites, and social media sites [20]. We became aware in our study that, when it comes to periodontitis and HIV, most websites dealing with this topic were sites from dental practices. This is probably due to the fact that oral health professionals are more aware of the increased prevalence of periodontitis in PLWH and deal more often with this issue.

For PLWH, HIV-related stigma could be an additional reason to look for online information. Because of this stigma, PLWH may avoid directly contacting oral health care providers about their HIV status [43]. Although medical and dental professionals’ knowledge of the transmission of HIV has increased, a recent study has shown that the stigma towards PLWH still persists in dental practice [41]. This persisting stigma results in a persistent reduction in the quality of health [42]. Therefore, the importance of correct Internet information about periodontal diseases is indispensable for PLWH. 

AI can also serve as a potential source of medical information, and its role in this field is rapidly evolving. Notably, in our study, we did not exclude any webpages created with the help of AI. Interestingly, none of the articles included in our study were reported to have been developed with the assistance of AI. For future studies, it would be intriguing to evaluate different AI tools to validate the information provided regarding HIV and periodontitis. Such an investigation could explore the accuracy, reliability, and quality of AI-generated content compared with human-created information. This would involve a detailed analysis of various AI platforms, assessing their ability to produce accurate and comprehensive medical information. However, conducting such a study would require a completely different methodology, including criteria for selecting AI tools, parameters for evaluating their output, and methods for comparing this output against established medical guidelines. The major strength of this study is that it is the first study to assess the quality of available Internet information about periodontitis in PLWH. Another strength of this study is that individuals restrict their search to the initial eight to ten results returned by a search engine [44,45] In this study, however, we expanded the number of included sites from ten to twenty results. A number of possible limitations need to be considered. A well-known limitation of this type of study is that, in real life, search results are heavily influenced by previous searches and the different Internet search algorithms that search engines use. Search engines have evolved significantly to adapt to the way individuals search for information. Google™, for example, uses complex algorithms that consider, amongst others, a user’s previous search history and location. In contrast, Bing™, another major search engine, focuses on user interface design and provides a more visually appealing experience, especially for image and video searches [46]. In essence, search engines are in a constant state of adapting to meet the ever-changing demands and preferences of users, which will also impact search results over time. Recognizing this limitation, and in order to alleviate some of this bias in our study, we cleared search histories and Internet caches (including cookies) and surfed in private mode. Secondly, in general, it is known that websites have a dynamic character, and their profile might change considerably in a very short period of time. Thus, the content and design might change, leading to different JAMA and/or FRES results. Therefore, Oklun et al. repeated their study after six months to see whether the quality of the Internet information found in their study was still up to date. The great similarity between the two search results showed that the quality of the information did not change quickly [37]. For this reason, we can assume that Internet information about periodontal diseases and HIV will not differ significantly. Thirdly, in this study, we only included websites in the English language, thus the results are limited to English sites. Searches in other languages may differ, limiting this study to a particular geography. However, assessing only websites in the English language is a common practice in studies assessing the quality of Internet medical information. Bizzi et al., Kanmaz et al., Kalichman et al., and other studies comparable to our study also restrained their searches to English websites [30,31,32,33]. Therefore, we expect that the included English websites are a valid representation of the worldwide situation. Lastly, specific characteristics of the website, such as font, layout, and illustrations used, are not included in the assessment. To the best of our knowledge, there are no tools yet available to assess the quality of layout, illustrations, and other features that could significantly affect readability. Consequently, to standardize reporting results of this category of research, a reporting guideline is eagerly awaited.

## 5. The Ideal Website

The quantitative characteristics of an ideal informational website should include accuracy, coverage, readability, and citations. Accurate content, derived from reputable medical sources and regularly updated, ensures reliability. The breadth and depth of information provided should cover a wide range of topics, with detailed explanations. Additionally, the inclusion of numerous citations from peer-reviewed articles and authoritative guidelines enhances the site’s credibility. Readability, measured by tools like the Flesch score, ensures that the content is easily understandable.

Qualitative characteristics should encompass authority, objectivity, clarity, user experience, and interactivity. The credibility of the information is bolstered by authorship from qualified medical professionals or reputable organizations. Objective unbiased information should be presented clearly, without jargon, ensuring that users can easily comprehend the material. A user-friendly experience, marked by intuitive navigation and a clean design, coupled with interactive tools such as symptom checkers and community I Cforums, enhances engagement. For PLWH and the general public, an ideal website should provide comprehensive up-to-date information, supported by credible sources, and designed for easy navigation and interactivity. Ensuring privacy and security of user data, along with accessible features for users with disabilities, would further contribute to the overall quality and reliability of the site.

## 6. Conclusions

Websites with information on periodontitis in PLWH require advanced reading skills and quality features are regularly not provided. Most websites do not go into details about the treatment of periodontitis and do not clearly explain the possible link between periodontitis and HIV. We recommend active involvement from healthcare professionals and PLWH in the development of online resources about periodontal diseases in this population. Ensuring accuracy and quality in these evidence-based materials is paramount, given the significance of reliable Internet information on the subject.

## Figures and Tables

**Figure 1 ijerph-21-00857-f001:**
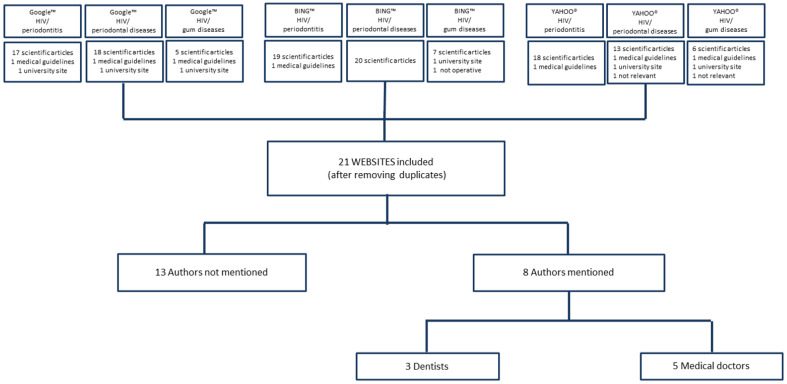
Flowchart of the selection process and author details of the websites.

**Figure 2 ijerph-21-00857-f002:**
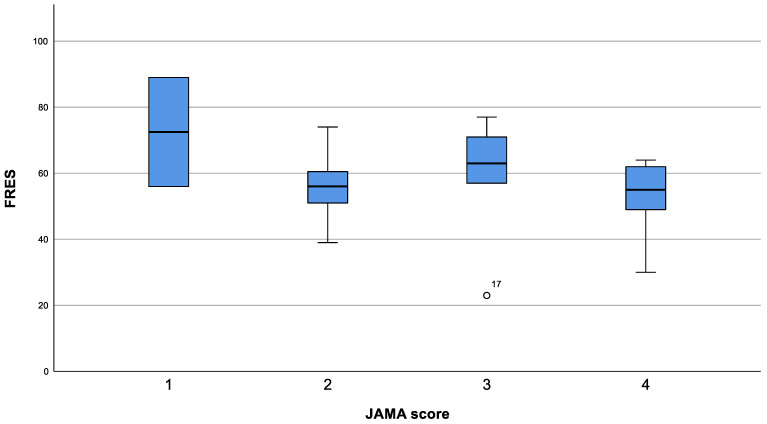
FRES distribution by JAMA.

**Table 1 ijerph-21-00857-t001:** Outcomes of the quality scores of the Internet information about periodontitis in PLWH.

FRES Mean (*SD*)		57.1 (15.0)
JAMA score mean (*SD*)		2.81 (1.0)
JAMA Total Score *N* (%)	1	2 (9.5)
	2	7 (33.3)
	3	5 (23.8)
	4	7 (33.3)
JAMA parameter 1: authorship *N* (%)		8 (38.1)
Background Author *N* (%)	Unknown	13 (61.9)
	Dentist	3 (14.3)
	Medical Doctor	5 (23.8)
JAMA parameter 2: attribution *N* (%)		11 (52.4)
JAMA parameter 3: disclosure ownership of the site *N* (%)		21 (100)
JAMA parameter 4: currency, date of publication *N* (%)		18 (85.7)
Origin	Government	5 (23.8)
	Dental organization	3 (14.3)
	HIV organization	5 (23.8)
	Non-profit organization	2 (9.5)
	News agencies	6 (28.6)
Type of Information on Site	Information about HIV and periodontal/oral diseases presented adequately	14 (66.7)
	General information about periodontal/oral diseases with some limited information about HIV	5 (23.8)
	Information only about periodontal/oral diseases	3 (14.3)
Quality of Advice	Good advice for selfcare and oral healthcare treatment	13 (61.9)
	Advice to see a dentist	1 (4.8)
	Advice for self-care	1 (4.8)
	No advice	5 (23.8)
Readership Targeted *N* (%)	People living with HIV	12 (57.1)
	General population	8 (38.1)
	Healthcare workers	1 (4.8)
Type Oral Diseases	All oral diseases	14 (66.7)
	Only periodontal diseases	5 (23.8)
	Caries	1 (4.8)
	Not named	1 (4.8)

**Table 2 ijerph-21-00857-t002:** Quality evaluation of included websites.

Sites (In Alphabetical Order)	FRES	JAMA
Six Dental and Oral Problems With HIV|myHIVteam	55	4
Eight Tips for Preventing HIV Gum, Teeth, and Oral Problems|myHIVteam	61	4
Dental Problems Associated With HIV/AIDS|Colgate^®^	54	2
Five Oral Manifestations Of HIV|Colgate^®^	60	2
HIV and Tooth Decay: The Link between Oral Health and HIV (yapmt.org)	57	3
HIV and Your Mouth|The Well Project	63	3
HIV Mouth Sores: What They Look Like and How to Treat Them (healthline.com)	64	4
HIV, AIDS, and Oral Health|MouthHealthy—Oral Health Information from the ADA	48	2
HIV/AIDS and Oral Health (webmd.com)	71	3
HIV-Related Mouth Sores: Symptoms and Treatments (medicalnewstoday.com)	52	4
Linear Gingival Erythema and HIV: Causes, Symptoms, and Treatment (healthline.com)	46	4
Linear Gingival Erythema and HIV: Symptoms, Treatment (verywellhealth.com)	30	4
Oral Health and HIV—Mississippi State Department of Health (ms.gov)	56	2
Oral Health and HIV (hrsa.gov)	56	1
Oral Health and HIV (ny.gov)	89	1
Oral Health and HIV|Positive Life NSW	63	4
Oral Health for People With HIV|HRSA	74	2
Oral Health: Tips for People With HIV (hrsa.gov)	77	3
Researchers Discover Link Between Gum Disease and HIV Progression—POZ	39	2
The Connection between AIDS and your Oral Health|Delta Dental (deltadentalins.com)	61	2
What Every HIV and AIDS Patient Should Know About Periodontal Disease|Ear, Nose, Throat, and Dental problems articles|Body and Health Conditions Center (SteadyHealth.com)	23	3
Mean (*SD*) *N* = 21 Median (IQR)	57.1 (15.0) 57.0 (50.0–63.5)	2.8 (1.0) 3.0 (2.0–4.0)

HIV: Human immunodeficiency virus; YAPMT: Yellowstone AIDS Project; AIDS: acquired immunodeficiency syndrome; ADA: American Dental Association; webMD: Website medical doctor; ms.gov: Mississippi government; HRSA: Health Resources & Services Administration; NSW: New South Wale; POZ: HIV positive.

## Data Availability

The datasets used and/or analyzed during the current study are available from the corresponding authors on reasonable request.

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
