# Peer review of "Quality Assessment of Internet Information Regarding Periodontitis in Persons Living with HIV"

_ijerph, 2024, doi:10.3390/ijerph21070857_

Round 1
Reviewer 1 Report
Comments and Suggestions for Authors
Overall assessment:
This article touches on an important issue on health literacy by reviewing available resources for a particularly vulnerable population group experiencing a specific health condition.
However, the rigor and scope of the methods used may not encompass the breadth of information in the web, particularly excluding an emerging source of information, AI. Also, to contextualize the issue and establish the significance of conducting this study, more studies that focus on health literacy and technology (or the lack thereof) must be presented (in lieu of describing the disease).
Below are comments and suggestions, and kindly pay close attention to comments re: methodology.
Specific suggestions and comments:
Line 30 - syntax issue, 'worldwide' placed after 39 million
Line 34 - punctuation issue, comorbidity-free
Line 38-9 - What is the prevalence rate? Quantify the extent of the difference.
Line 43 - syntax issue, remove "A" before HIV-related
Line 51 - punctuation issue, Two periods
Line 52 - appropriate citation style needed (website)
Line 58 - Define or specify quan and qual characteristics. What makes an ideal informational website? What parameters should PLWH or the general public be looking for?
Line 63 - Is cross-sectional internet search a design? I believe this is more of a method. I suggest describing the design as a cross-sectional study of internet sites relevant to periodontal disease and HIV.
Line 78 - suggest a better term for "surfed"
Line 79 - Explain why only the first 20 results are included, knowing the possibility that "non-sponsored" websites may not show up within those limits.
Line 101-2 - What was the basis for selection of web content for copy-pasting and analysis? Websites can have multiple pages and information so this has to be clear.
Line 134-5 - The mention of multiple statistical analyses does not help in the clarity on how they are appropriate for the data collected. Furthermore, the biggest issue in this study is that there is no evidence of such analyses reflected in the results, being that they are primarily descriptive (means, SD, etc.). If indeed tests like t-test and ANOVA were conducted, what were the hypotheses being tested?
Line 228-30 - When were these studies conducted? It is important to mention the year that they were conducted because as you have said, most information in the internet is constantly evolving.
Line 252 - Missing word: Organizations "such" as...
Line 258 - word change: easier to understand
Line 296, - word choice, change seriously to significantly
Line 310-2 - Would this not pull the readability score lower, that is, by going into these details?
Comments on the Quality of English Language
Overall, quality of English language is fine. There may be some word choice issues which can be improved to enhance readability. In addition, word placement (i.e., syntax) in some sentences may need to be reviewed.
Author Response
Please see the attechment

Reviewer 2 Report
Comments and Suggestions for Authors
Statistical terms such as p values always begin with a lowercase, italicized letter.
Authors should revise for some typos.
I recommend mentioning in the discussion section not only the limitations but also the strengths of this study.
Reviewer 3 Report
Comments and Suggestions for Authors
This is an interesting computational study, which assessed the Internet information about periodontitis in persons with HIV. The authors found the current English website information on periodontitis in HIV (+) persons require advanced reading skills and the quality features are regularly not provided. The major and minor comments are listed as follows:
Major comments:
1. It is important about “correctness of information” provided to Internet readers. It would have been better if periodontist(s) had participated in the research.
2. About section 2.3.3., the authors did not provide further description about the results of assessing these criteria.
Minor comments:
1. There should be a space before an opening bracket symbol.
2. Line 51, Page 2: “insights. [17].” has an extra period symbol.
3. The formats for the acronym of “Flesch Reading Ease Score” are inconsistent.
4. Line 126-142, Page 3: The decimal symbols for P values are inconsistent, nor as thousand separators
5. Table 2, Figure 2: “Frescore” is different from “FREScore” and ”FRES score” regarding formats.
6. References: The formats of listed references are inconsistent, especially about upper- and lower-cases. The authors should carefully revise each reference.
7. Reference #4: “Hiv” should change to “HIV”.
8. Reference#22: There is an extra space after “seeking”.
9. Reference #40: It should be “J Forensic Sci”. There is an extra “i”.
Comments on the Quality of English LanguageOnly minor polishing is required.
Reviewer 4 Report
Comments and Suggestions for Authors
The authors have assessed the quality of Internet Information Regarding Periodontitis in Persons Living with HIV. Although I fin d this paper interesting, I have a few queries and suggestions for improvmenet
1. Please elaborate on the methodology section: Add a search string for MESH terms and keywords used to search the relevant articles. The authors have not included important sources of guidelines for infectious diseases, such as the CDC guidelines, WHO guidelines, and NHS guidelines. Please add the papers and information available from these sources. Additionally, the source should provide how they screened the article and which articles were excluded.
2. Internet sources is a broad term, which may include, journal articles, editorials, newspapers, guidelines by societies, and public blogs. Have the authors considered all these webpages and are there specific criteria for selection? If so, please mention the criteria. I suggest including all and analyzing the data in different databases and then analyzing which source of evidence should be considered to be accurate and of the best quality.
3. Please write the results more systematically, with headings and subheadings. You may divide the results into sections: quantity of information available on each aspect of HIV: for example, how is the quality of information with regards to transmission; with regards to diagnosis, or its management. Which source has the best quality in each of these aspects?
4. Performing a subgroup analysis among different sources is recommended
5. The limitations and use of this evidence for the public.
Comments on the Quality of English Language
Can be improved.
Round 2
Reviewer 1 Report
Comments and Suggestions for Authors
Thank you for addressing my initial review. This is a very much improved version.